# Abnormalities of External Jugular Veins in Bulldogs with Pulmonary Valve Stenosis

**DOI:** 10.3390/vetsci9070359

**Published:** 2022-07-15

**Authors:** Marta Croce, Tommaso Vezzosi, Federica Marchesotti, Valentina Patata, Martina Bini, Giuseppe Lacava, Luigi Venco, Oriol Domenech

**Affiliations:** 1Anicura Istituto Veterinario Novara, 28060 Novara, Italy; federica.marchesotti@gmail.com (F.M.); valentinapatata89l@gmail.com (V.P.); binimartina93@gmail.com (M.B.); giuseppe.lacava@anicura.it (G.L.); odomenech1973@gmail.com (O.D.); 2Department of Veterinary Sciences, University of Pisa, 56122 Pisa, Italy; tommaso.vezzosi86@gmail.com; 3Veterinary Hospital City of Pavia, 27100 Pavia, Italy; luigivenco@libero.it

**Keywords:** cardiology, interventional cardiology, pulmonic stenosis, central venous catheterization

## Abstract

**Simple Summary:**

This case series is about two French bulldogs and one English bulldog affected by pulmonary valve stenosis that were diagnosed with different abnormalities of the external jugular veins (EJV) before a pulmonary balloon valvuloplasty. The EJV abnormalities encountered were severe hypoplasia of both external jugular veins and right external jugular vein absence, associated with persistent left cranial vena cava. All cases were diagnosed through computed tomography and vascular ultrasound. The aim of this case series is to describe external jugular veins abnormalities that can affect the decision-making process regarding central venous catheterization needed to perform procedures such as pulmonary balloon valvuloplasty or central catheter placement. Based on the results of the present case series, anomalies of the external jugular veins should be considered in French and English bulldogs when the external jugular veins cannot be clinically identified or when echocardiography shows coronary sinus dilation. Vascular ultrasound or computed tomography may help identify jugular venous anomalies and should be considered in the preoperative evaluation of bulldogs that need to undergo interventional procedures requiring transjugular catheterization.

**Abstract:**

Two French bulldogs and one English bulldog affected by pulmonary valve stenosis and referred for pulmonary balloon valvuloplasty were diagnosed with different abnormalities of the external jugular veins. The diagnosis included unilateral absence of the right external jugular vein and bilateral hypoplasia of the external jugular veins, associated with persistent left cranial vena cava. Vascular ultrasound and computed tomography were used for the diagnosis. Jugular vein anomalies can affect decision-making regarding the central venous catheterization needed to perform procedures such as pulmonary balloon valvuloplasty or central catheter placement. Based on the results of the present case series, anomalies of external jugular veins should be considered in French and English bulldogs when the external jugular veins cannot be palpated or when echocardiography shows coronary sinus dilation. Vascular ultrasound or computed tomography may help identify jugular venous anomalies and should be considered in the preoperative evaluation of bulldogs that need to undergo interventional procedures requiring transjugular catheterization.

## 1. Introduction

Pulmonary valve stenosis (PS) is one of the most frequent congenital heart diseases in dogs [1], especially in English and French bulldogs [2,3]. These breeds are also frequently affected by coronary artery anomalies [4,5,6,7,8,9]. Pulmonary balloon valvuloplasty (PBV) has become the elective treatment for dogs with severe forms of PS [3,10], with a low rate of complications and a high rate of success in reducing the transvalvular pressure gradient [11].

Conducting this intervention entails catheterizing an external jugular vein (EJV) in order to perform a right heart catheterization and reach the anomalous valve with the balloon [12]. PBV can also be carried out through the femoral veins, but the EJVs are larger than femoral veins; thus, the vascular access is usually easier [13]. Moreover, if the catheter is inserted through the cranial vena cava via the EJV, there is a natural tendency for catheters to pass through the right atrium and tricuspid valve and then directly into the right ventricular outflow tract [13]. In our experience and in the veterinary literature, it is easier to manipulate the catheter entering through the EJVs than through the femoral veins [3]. It is, thus, very important to evaluate the presence as well as the conditions of the EJV before right cardiac catheterization. Anomalies of the EJV in dogs have been reported in only one study on English bulldogs [14].

Therefore, the aim of this case series was to describe three bulldogs affected by PS and concomitant EJV abnormalities, which entails rethinking the vascular access needed to perform the PBV.

## 2. Case Presentation

### 2.1. Case 1

An 18-month-old, female French bulldog weighing 11 kg was referred to the Cardiology Department of the Anicura Istituto Veterinario di Novara for the management of a severe PS. Cardiac auscultation revealed a grade 4/6 pansystolic ejection murmur over the left heart base. Examination of the neck and jugular veins revealed the presence of the left EJV while the right EJV was not clearly identified. The remaining physical examination was unremarkable.

The consultation included a complete echocardiographic examination using a GE Vivid Iq with a 2–8 MHZ phased array probe, which showed severe PS with a pulmonary peak systolic pressure gradient of 143 mmHg, right ventricular hypertrophy, and right atrial enlargement. PBV was, thus, recommended. Blood tests showed no abnormalities.

Before the PBV, a vascular ultrasound of the neck was performed using a GE Logiq S8 with a 10 MHZ linear probe, which confirmed the absence of the right EJV. Thus, the PBV was successfully performed using a left EJV access without complications (Figure 1).

### 2.2. Case 2

A one-year-old, male French bulldog weighing 9.4 kg was referred to the Cardiology Department of the Anicura Istituto Veterinario di Novara for the management of PS. On presentation, a grade 4/6 pansystolic ejection murmur was auscultated over the left heart base. Similar to Case 1, the examination of the neck and EJVs revealed the presence of the left EJV while the right EJV was not identified. The remaining physical examination was unremarkable.

A complete echocardiographic examination confirmed severe PS with a peak systolic pressure gradient of 121 mmHg. Right ventricular hypertrophy and dilation were evident as were right atrial enlargement and dilation of the coronary sinus (Figure 2). Pulmonary balloon valvuloplasty was recommended after having performed an ECG-gated computed tomography angiography (CTA) to evaluate the coronary arteries, the dilated coronary sinus, and the suspected absence of the right EJV. Blood tests showed unremarkable results.

The CTA, performed using a GE Optima 660 (64 slices), revealed persistent left cranial vena cava and no coronary artery anomalies; it also confirmed the absence of the right EJV.

Given the absence of the right EJV and the persistence of the left cranial vena cava, the PBV was successfully performed by vascular access through the right femoral vein without complications (Figure 3).

### 2.3. Case 3

A one-year-old, female English bulldog weighing 16 kg was referred to the Cardiology Department of the Anicura Istituto Veterinario di Novara for the management of PS. Cardiac auscultation revealed a 4/6 pansystolic ejection murmur over the left heart base. Examination of the neck and jugular veins did not clearly identify external jugular veins. The rest of the physical examination was unremarkable.

Complete echocardiographic examination confirmed the presence of severe PS with a pulmonary peak systolic gradient of 81 mmHg. Right ventricular hypertrophy and right atrial enlargement were evident. An ECG-gated CTA was performed to rule out the possible presence of coronary artery abnormalities prior to the BPV. Blood tests showed unremarkable results. The CTA revealed severe bilateral hypoplasia of the EJVs with a secondary compensatory increase in the diameter of internal jugular veins (Figure 4). No coronary artery anomalies were observed.

Given the impossibility of performing the vascular access through the EJVs, the PBV was successfully performed using vascular access from the right femoral vein without complications.

## 3. Discussion

This paper reports two cases of congenital unilateral absence of an EJV in two French bulldogs and one case of severe bilateral hypoplasia of the EJVs in one English bulldog, all of which were referred for PBV.

To the best of our knowledge, although the absence of an EJV has already been reported for two English bulldogs and one Jack Russel terrier [14], it has never been reported for French bulldogs. Moreover, severe hypoplasia of both EJVs has never been described in dogs. These vascular abnormalities have a clinical impact on the decision-making process regarding vascular accesses in dogs that need to undergo transvenous catheter interventions.

The EJVs arise from the union of the linguofacial and the maxillary veins. The EJVs end in the subclavian veins, which form the brachiocephalic veins [15]. Both brachiocephalic veins then merge to form the rightward cranial vena cava, which is an unpaired vessel that lies in the cranial mediastinum ventral to the trachea and empties into the cranial part of the right atrium [15]. The internal jugular veins arise from the confluence of the petrosal and sigmoid sinuses at the level of the tympano-occipital fissure and continue caudoventrally near the internal and common carotid arteries, deeper than the EJVs. In dogs, the EJVs are larger than the internal ones and represent the dominant pathway for the venous return from the head [15]. They are also the most frequently used venous vascular access points in the neck for right heart catheterization during interventional procedures in dogs (e.g., PBV, pacemaker implantation).

Since the veins of the head and neck have a complex developmental pattern, variations in formation and drainage as well as vascular malformations can occur; thus, causes of EJV abnormalities are considered to be embryogenic [16,17]. There are several vascular malformation classification schemes in the human literature, but the most frequently cited is the Hamburg classification [18]. This system considers anatomical, histological, and pathophysiological features of congenital vascular malformations and introduces embryological aspects, subdividing malformations into the extratruncular or truncular form, based upon the time of developmental abnormality during the embryogenesis [19].

The cases reported here referred to truncular venous malformations, which are an anomaly of the tubular formed vessels of the EJVs. In particular, the absence of the EJV observed in our two French bulldogs represents a truncular aplasia while the hypoplasia of the EJVs is an example of truncular hypoplasia as identified in our English bulldog. Truncular venous aplasia and hypoplasia can result in compensatory dilation of collateral veins [20], such as the internal jugular veins, as observed in our English bulldog case. Truncular venous aplasia of the cervical venous system has been reported in humans, cats, and English bulldogs [11,21] while, to the best of our knowledge, hypoplasia of the EJVs has never been previously reported in humans, cats, or dogs.

When compared to humans, EJV anomalies in people are often discovered at postmortem examination because the internal jugular veins have a more relevant clinical aspect since they are larger and used for central venous catheterization [14,22,23,24,25,26].

Another anomaly of the venous system in dogs is the persistence of the left cranial vena cava, which is a rarely recognized vascular anomaly comprising less than 5% of congenital cardiovascular defects diagnosed in dogs [11,27]. The persistent left cranial vena cava emptying into the right atrium through the coronary venous sinus usually has no hemodynamic consequences, but it commonly accompanies other congenital cardiovascular defects, such as PS, and may complicate cardiac catheterization [27]. The dilation of the coronary sinus is a consequence of this anomaly and seems to predispose the patient to cardiac arrhythmias due to stretching of the atrioventricular node and the bundle of His in people [28]. Using the left EJV in the case of a persistent left cranial vena cava causes catheters and guidewires to enter the heart through the coronary sinus into the caudal aspect of the right atrium, leading to complications in the procedure and increasing procedural risks due to possible fatal arrhythmias or cardiac perforation [27].

The dilation of the coronary sinus detected during preprocedural echocardiography, as occurred in Case 2, is suggestive of a persistent left cranial vena cava. However, it is not specific for this condition, and the definitive diagnosis of this anomaly requires advanced diagnostic imaging such as CTA [14]. Similar to our case series, a previous study [14] reported EJV anomalies as a limitation that should be considered for central venous catheterization, and all cardiologists performing interventional procedures should be aware of these possible vascular anomalies.

Cervical venous anomalies can be diagnosed with a variety of imaging technologies, including CTA, magnetic resonance imaging, vascular ultrasound, and fluoroscopy. CTA and magnetic resonance imaging are the gold standard diagnostic tests for the characterization of vascular anomalies and allow the detection of other cardiac or vascular concurrent congenital anomalies. However, expensive instrumentation and general anesthesia may represent limitations for these procedures. Although vascular ultrasound of the neck may provide less overall detailed information on the vessel course compared to the CTA, it still helps in detecting the presence or absence of the EJV [14,21], especially in obese dogs or those patients, such as English bulldogs, with large neck conformation and prominent skin folds that do not allow an accurate jugular vein detection through clinical examination. Furthermore, with the growing number of interventional procedures and central venous catheterizations in dogs and cats, the presence of the EJVs should be always evaluated [14,21].

In conclusion, based on the abnormalities of the EJVs described in the present case series in two French bulldogs and one English bulldog, it is recommended to always rule out vascular anomalies before central venous catheterization. Vascular abnormalities might be assessed through ultrasound examination or advanced diagnostic imaging modalities. However, considering the high prevalence of concurrent cardio-vascular anomalies in these breeds, CTA should be considered as the most appropriate diagnostic tool.

## Figures and Tables

**Figure 1 vetsci-09-00359-f001:**
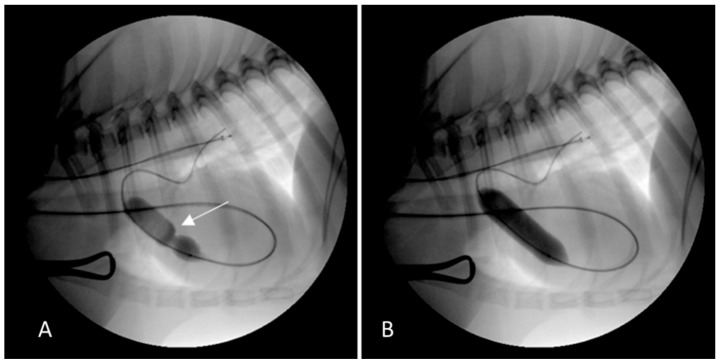
Fluoroscopic images of the pulmonary balloon valvuloplasty in Case 1 performed using vascular access through the left external jugular vein access. (**A**) The balloon was partially inflated across the pulmonary valve stenosis. The stenosis was identified by the waist, which was seen as the balloon was inflated (white arrow). The guidewire passed into the cranial vena cava, through the right atrium and the right ventricle, and out into the pulmonary artery. (**B**) The balloon was completely inflated causing the waist to abruptly resolve.

**Figure 2 vetsci-09-00359-f002:**
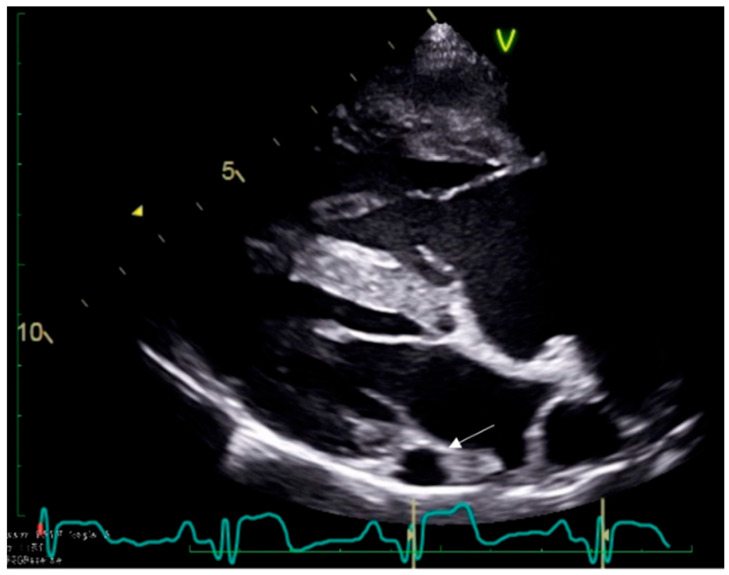
Echocardiographic right parasternal long axis view showing right ventricular hypertrophy and dilation, right atrial enlargement, and dilation of the coronary sinus (white arrow) in Case 2.

**Figure 3 vetsci-09-00359-f003:**
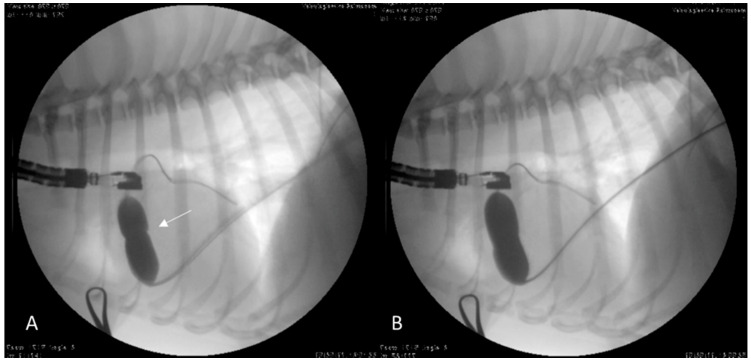
Fluoroscopic images of the pulmonary balloon valvuloplasty in Case 2 performed using vascular access through the right femoral vein. (**A**) The balloon was partially inflated across the pulmonary valve stenosis. The stenosis was identified by the waist, which was seen as the balloon was inflated (white arrow). The guidewire passed into the caudal vena cava, through the right atrium and right ventricle, and out into the pulmonary artery. (**B**) The balloon was completely inflated causing the waist to abruptly reduce its size.

**Figure 4 vetsci-09-00359-f004:**
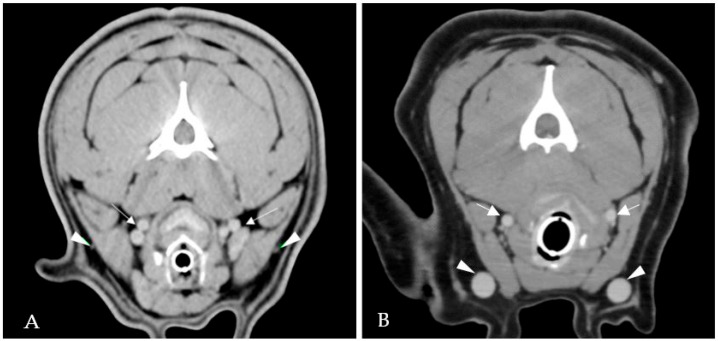
ECG-gated computed tomography angiography in Case 3. Severe bilateral hypoplasia of the external jugular veins (white arrowheads) with a secondary compensatory increase in the diameter of both internal jugular veins (white arrows) (**A**). ECG-gated computed tomography angiography in a dog with normal cervical venous system. External jugular veins (white arrowheads) and internal jugular veins (white arrows) (**B**).

## Data Availability

The data presented in this study are available on request from the corresponding author. The data are not publicly available due to privacy.

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
