# Peer review of "Abnormalities of External Jugular Veins in Bulldogs with Pulmonary Valve Stenosis"

_vetsci, 2022, doi:10.3390/vetsci9070359_

Round 1

Reviewer 1 Report

Accepted for publication with a minor revision of  English language

Reviewer 2 Report

In this study, the authors reported 3 cases of external jugular vein (EJV) abnormalities in Bulldogs with pulmonary valve stenosis. The authors confirmed the EJV abnormalities in bulldogs visited for treatment of pulmonary valve stenosis by imaging modality (ultrasonography, computed tomography). Through this, the authors suggest that vascular ultrasound or computed tomography should be considered to identify EJV anomalies when transjugular catheterization is required (especially in bulldogs). Overall, this is an interesting case report that will help veterinary clinicians. However, it seems that the following points need to be improved to more clearly convey the authors' point of view.

1. The authors' figures, with the exception of figure 4, relate to the heart or approach to pulmonary stenosis rather than to explain the abnormality of the jugular vein. Three cases should be presented through images of the jugular vein observed by the authors (comparison of left and right if possible).

2. The authors' narratives largely remain "observational". What are the main causes of jugular vein abnormalities in Bulldog breeds? Or how does this feature in other species? How about humans? A little more comparative explanation seems to be necessary.

Reviewer 3 Report

The current case series described three cases of congenital anomalies in three bulldogs: two cases of congenital unilateral absence of an EJV in two French Bulldogs and one case of severe bilateral hypoplasia of the EJVs in one English Bulldog. All cases were referred for pulmonary balloon valvuloplasty (PBV). The cases were adequately described. I recommend considering this article for publication in Veterinary Sciences after taking into account the following remarks:

1. The heading of section two should become (2.Case presentation) instead of (3.Results) since it merges the methodology and findings in one section.

2. The authors have conducted  vascular ultrasound of the neck prior to PBV. In my opinion, a representative image of the ultrasound findings should be included.

3. In figure 4, if possible, It would be more valuable and clear if an image from normal dog is included.

Round 2

Reviewer 2 Report

I appreciate the great efforts that the authors have made in response to my questions and concerns. Although, it is unfortunate to not see the images of Cases 1 and 2, the authors' claims and supporting results are considered to be sufficiently interesting for clinicians.